# Meeting at the Crossroad between Obesity and Hepatic Carcinogenesis: Unique Pathophysiological Pathways Raise Expectations for Innovative Therapeutic Approaches

**DOI:** 10.3390/ijms241914704

**Published:** 2023-09-28

**Authors:** Konstantinos Arvanitakis, Stavros P. Papadakos, Vasileios Lekakis, Theocharis Koufakis, Ioannis G. Lempesis, Eleni Papantoniou, Georgios Kalopitas, Vasiliki E. Georgakopoulou, Ioanna E. Stergiou, Stamatios Theocharis, Georgios Germanidis

**Affiliations:** 1First Department of Internal Medicine, AHEPA University Hospital, Aristotle University of Thessaloniki, 54636 Thessaloniki, Greece; arvanitak@auth.gr (K.A.); elenispap@yahoo.gr (E.P.); gekalopi@auth.gr (G.K.); 2Basic and Translational Research Unit (BTRU), Special Unit for Biomedical Research and Education (BRESU), Faculty of Health Sciences, School of Medicine, Aristotle University of Thessaloniki, 54636 Thessaloniki, Greece; 3First Department of Pathology, School of Medicine, National and Kapodistrian University of Athens, 11527 Athens, Greece; stpap@med.uoa.gr (S.P.P.); stamtheo@med.uoa.gr (S.T.); 4Department of Gastroenterology, School of Medicine, National and Kapodistrian University of Athens, Laiko General Hospital, 11527 Athens, Greece; lekakis.vas@gmail.com; 5Division of Endocrinology and Metabolism and Diabetes Centre, First Department of Internal Medicine, Medical School, AHEPA University Hospital, Aristotle University of Thessaloniki, 54636 Thessaloniki, Greece; thkoyfak@auth.gr; 6Institute of Metabolism and Systems Research (IMSR), College of Medical and Dental Sciences, University of Birmingham, Birmingham B15 2TT, UK; lemp.ioan@gmail.com; 7Department of Pulmonology, Laiko General Hospital, 11527 Athens, Greece; vaso_georgakopoulou@hotmail.com; 8Pathophysiology Department, School of Medicine, National and Kapodistrian University of Athens, 11527 Athens, Greece; stergiouioanna@hotmail.com

**Keywords:** obesity, metabolic dysfunction-associated fatty liver disease, hepatocellular carcinoma

## Abstract

The escalating global prevalence of obesity and its intricate association with the development of hepatocellular carcinoma (HCC) pose a substantial challenge to public health. Obesity, acknowledged as a pervasive epidemic, is linked to an array of chronic diseases, including HCC, catalyzing the need for a comprehensive understanding of its molecular underpinnings. Notably, HCC has emerged as a leading malignancy with rising incidence and mortality. The transition from viral etiologies to the prominence of metabolic dysfunction-associated fatty liver disease (MAFLD)-related HCC underscores the urgent need to explore the intricate molecular pathways linking obesity and hepatic carcinogenesis. This review delves into the interwoven landscape of molecular carcinogenesis in the context of obesity-driven HCC while also navigating using the current therapeutic strategies and future prospects for combating obesity-related HCC. We underscore the pivotal role of obesity as a risk factor and propose an integrated approach encompassing lifestyle interventions, pharmacotherapy, and the exploration of emerging targeted therapies. As the obesity-HCC nexus continues to challenge healthcare systems globally, a comprehensive understanding of the intricate molecular mechanisms and innovative therapeutic strategies is imperative to alleviate the rising burden of this dual menace.

## 1. Introduction

Obesity has been officially recognized by the World Health Organization (WHO) as an epidemic [1]. The Framingham Heart Study recognized obesity as an independent risk factor for cardiovascular disease [2], contributing to the conceptualization of obesity as a morbidity factor. Since then, tons of evidence has linked obesity with a plethora of disorders, including hypertension, hyperlipidemia, insulin resistance, type 2 diabetes mellitus (T2DM), coronary and cerebral artery disease, gallbladder disease, metabolic dysfunction-associated fatty liver disease (MAFLD), obstructive sleep apnea, osteoarthritis, and malignancy, that contribute to increased morbidity and mortality [3,4]. Increased economic burden from obesity-related diseases for public health systems has been documented in a multitude of clinical studies [5,6,7]. Evidence suggests that irrespective of the accessibility of a health system to the public, the geographic location, or the studied era, obesity increases significantly both direct (hospitalized patients, outpatients, and drug prescription) [5,7] and indirect costs [6]. People with obesity are classified according to their body mass index (BMI) into four categories: overweight (25–29.9), obesity class I (30–34.9), obesity class II (35–39.9), obesity class III (morbid obesity—above 40) [3]. The American College of Cardiology and the American Heart Association, in order to confront the rising wave of obesity [8], have published guidelines for its management in adults [4]. Recent data regarding global trends in obesity are disheartening [9]. Over the last four decades, an increase in the mean BMI has been documented in both sexes, from 21.7 kg/m^2^ and 22.1 kg/m^2^ to 24.2 kg/m^2^ and 24.4 kg/m^2^ for men and women, respectively [9]. In adulthood, it seems that in wealth-producing countries, the increase in BMI has reached a plateau in contrast with low-income countries, such as certain parts of Africa and Asia, in which an increasing trend is being reported [8]. A recent systematic review involving 63 studies and 1,201,807 individuals revealed a global MAFLD incidence of 4613 cases per 100,000 person-years. The incidence rate has tripled between 2000 and 2015, with higher rates in men and those who are overweight/obese, emphasizing the need for public health interventions targeting at-risk populations as the prevalence of MAFLD continues to rise [10].

Hepatocellular carcinoma (HCC) comprises the seventh most frequently diagnosed malignancy and the second most common cause of cancer-related death worldwide [11]. MAFLD, the term proposed in 2020 to refer to fatty liver disease related to systemic metabolic dysregulation, is the most rapidly rising cause of HCC in Western populations since vaccination programs (Hepatitis B) and medications (Hepatitis C) have limited the expansion of viral hepatitis [12,13]. The most significant difference between nonalcoholic fatty liver disease (NAFLD) and the diagnosis of MAFLD is the removal of exclusion of concurrent liver disease to entertain the diagnosis. In more detail, the criterion for the diagnosis of MAFLD requires the presence of hepatic steatosis of ≥5% and identifies metabolic dysregulatory factors as a pre-requisite, which are T2DM, overweight/obesity by BMI classifications, and the presence of at least 2 out of 7 metabolic risk abnormalities, which include waist circumference, blood pressure, plasma triglycerides, plasma high-density lipoprotein-cholesterol, prediabetes, homeostasis model assessment of insulin resistance score, and plasma high sensitivity C-reactive protein [14]. The combination of hepatic steatosis with one of the aforementioned metabolic risk stratifications leads to the diagnosis of MAFLD [15]. The introduction of combined immunotherapy with angiogenic inhibition (atezolizumab plus bevacizumab) has revolutionized the systemic treatment of unresectable HCC, prolonging the median overall survival (OS) from less than six months to 19.2 months [16,17]. Obesity comprises a well-described etiologic factor of cancer. In fact, it is documented that each 5-unit rise in BMI increases approximately by 6% the risk of cancer [18]. Moreover, recent evidence suggests that metabolically unhealthy individuals have a 22% higher probability of liver cancer, with central obesity being a contributing factor [19]. These data are in accordance with evidence from a meta-analysis of prospective studies, which showed that individuals with overweight and obesity had a 48% and 83% greater risk for primary liver malignancy, respectively, in comparison with normal weight population [20]. It was estimated that a 5-unit increase in BMI, raised the relative risk for HCC by 39% [21]. T2DM is predominantly associated with cardiovascular complications [22], while in patients with T2DM, the prevalence of MAFLD, severe fibrosis, and cirrhosis have been estimated to be 65%, 14%, and 6%, respectively, with obesity and insulin administration being predictors of fibrosis [23]. A recent meta-analysis of six international cohorts demonstrated that patients with MAFLD and T2DM face a significantly increased risk of hepatic decompensation and HCC as compared to those without diabetes [24]. More specifically, participants with T2DM had a significantly higher risk of hepatic decompensation (at 5 years: 13.85% with T2DM vs. 3.95% without T2DM) and development of HCC (at 5 years: 3.68% with T2DM vs. 0.44% without T2DM) [24]. HCC could arise either in the context of MAFLD-related cirrhosis with an estimated incidence of 10.6 cases per 1000 patient years [25], or among MAFLD/metabolic dysfunction-associated steatohepatitis (MASH) patients with an estimated incidence varying from 0.08 to 0.62 per 1000 patient-years [26,27,28]. In fact, among non-cirrhotic individuals with chronic liver disease, MASH patients have the highest probability to develop HCC (OR 2.61, 95% CI 1.27–5.35, *p* = 0.009) [29]. An estimated 12% to 30% of MAFLD-related HCC arises without the presence of cirrhosis [16,30].

The aim of this narrative review is to shed light on the specific pathophysiologic mechanisms leading to HCC in the context of obesity, highlight current evidence-based and innovative therapeutic approaches, as well as to address future research challenges and opportunities in the field of obesity-related HCC.

## 2. Obesity and MAFLD: Molecular Pathways and Pathogenesis

The criteria for a positive diagnosis of MAFLD are based on histological (biopsy), imaging, or blood biomarker evidence of hepatic steatosis, in addition to one of the following three criteria, namely overweight/obesity, presence of T2DM, or evidence of metabolic dysregulation [14,15]. Then, MAFLD may progress to MASH, which is characterized by lobular or portal inflammation and damage to hepatocytes (ballooning), ultimately leading to cirrhosis, hepatic fibrosis, and/or HCC.

MAFLD, alongside its more serious manifestation, MASH, constitutes a major health burden, as results from a recent meta-analysis demonstrated that MAFLD and MASH among obese individuals have a global prevalence of 75% and 33%, respectively [31], with an increasing incidence due to the rise in T2DM and metabolic syndrome [32,33]. The progression from MASH to MASH-related HCC (~2% of cases per year) is influenced by a variety of factors, including the hepatic tissue and immune microenvironment, genetic polymorphisms, mainly patatin-like phospholipase domain containing 3 (PNPLA3) variant (rs738409; I148M) [34], and dysbiosis of the gut microbiota [35], while MASH-related HCC has unique molecular and immune traits as compared to other aetiologies of HCC. MASH is a disease that is histologically characterized by fatty changes with evidence of lobular hepatitis, focal necroses with mixed inflammatory infiltrates, and the presence of Mallory bodies, while it often advances to cirrhosis, predisposing to HCC. In more detail, in obesity, adipose tissue is dysfunctional and lacks the ability to store excess energy, leading to increased lipolysis and insulin resistance [36]. As a result, circulating free fatty acids (FFAs) and leptin increase while adiponectin decreases, resulting in intrahepatic fat accumulation, further potentiating de novo lipogenesis associated with the usual high-calorie intake observed in obese individuals. Adipose tissue is also further infiltrated by immune cells that produce cytokines and interleukins, contributing to a chronic low-grade intrahepatic inflammatory process, while mitochondrial defects, endoplasmic reticulum (ER) and oxidative stress link lipotoxicity and glucotoxicity with MASH [37,38]. In the setting of chronic low-grade intrahepatic inflammation, characterized by the activation of Kupffer cells, dendritic cells and hepatic stellate cells (HSCs), the liver is progressively infiltrated by immune cells, including neutrophils, T-lymphocytes and macrophages, and fibrogenesis ensues, which may result in cirrhosis, hindering the replacement of hepatocytes subjected to cell death or apoptosis, leading to a microenvironment that favors HCC development [39] (Figure 1).

A significant body of evidence has demonstrated that the accumulation of triglycerides in the liver is not inherently detrimental but rather serves as an adaptive mechanism to enhance the influx of FFAs [40]. However, the development and advancement of hepatocellular injury, inflammation, activation of hepatic stellate cells, and accumulation of extracellular matrix (ECM), collectively defining the phenotype of MASH, are primarily attributed to lipotoxicity [41]. In more detail, six-month treatment with omega-3 n-6/n-3 polyunsaturated fatty acids (PUFAs) greatly improved hepatic proteomic and plasma lipidomic markers of lipogenesis, endoplasmic reticulum stress, and mitochondrial functions in patients with MASH [42]. An excessive influx of fatty acids into hepatocytes contributes to the promotion of oxidative stress, which, in turn, leads to the development of insulin resistance (IR) and hinders the process of autophagy, resulting in the activation of apoptotic cascades, eventually leading to tissue damage and inflammation [43,44].

In animal models with obesity and T2DM, hyperinsulinemia promotes profibrogenic signals in HSCs via the stimulation of connective tissue growth factor (CTGF) mRNA transcription, either directly or as a co-factor of transforming growth factor-beta (TGF-β) [45], while results from another study demonstrated that TGF-β-mediated CTGF expression in HSCs requires signal transducer and activator of transcription 3 (STAT3) signaling activation [46]. Furthermore, patients with T2DM and MAFLD commonly experience hyperglycemia, which leads to the rapid formation of advanced glycation end products via non-enzymatic glycation of proteins in the bloodstream [47]. A study conducted by Jiang JX et al. demonstrated that these products elicit the fibrogenic activity and the production of reactive oxygen species (ROS) in MASH by influencing the activity of the tumor necrosis factor-alpha (TNF-α) converting enzyme (TACE) via NOX2 activation, as well as by downregulating the Sirt1/Timp3 pathways [48]. All these pathways demonstrate the underlying reasons for the enhanced progression of liver fibrosis and disease complications observed in patients with MAFLD and T2DM.

Furthermore, a double-blind, randomized clinical trial by Šmíd et al. demonstrated that twelve months of n-3-PUFA treatment of patients with MAFLD was associated with a significant decrease in gamma-glutamyl transferase (γGT) activity, liver steatosis reduction in those who lost weight, and beneficial changes in the plasma lipid profile, with n-3-PUFA-enriched triacylglycerols and phospholipids being the most expressed lipid signatures [49]. Recent research findings suggest that hepatocyte-derived extracellular vesicles as saturated fatty acid (SFA)-enriched transporters target macrophages and Kupffer cells, activating a Toll-like receptor-4 (TLR4)-mediated pro-inflammatory response enough to induce hepatocyte insulin resistance [50]. These mechanisms include the activation of death receptor signaling, the induction of ER stress leading to mitochondrial apoptosis, the stimulation of TLR4, the activation of inflammasomes, as well as the inhibition of autophagy.

## 3. The Progression of MAFLD to HCC

The complex association between MAFLD and tumor development is mediated by the interaction of multiple pathogenic pathways [51]. Chronic low-grade inflammation and IR, via the insulin-like growth factor-1 (IGF-1) axis activation, contribute to the creation of a microenvironment favoring carcinogenesis [52]. In addition, the presence of dysfunctional adipose tissue characterized by reduced adiponectin and elevated leptin secretion plays a crucial role in promoting cellular proliferation and angiogenesis [53,54]. Lastly, the presence of gut dysbiosis and increased intestinal permeability induce the translocation of bacterial metabolites, triggering TLR activation, which in turn promotes tumorigenesis by reducing the secretion of interleukin (IL)-18 and enhancing IL-6 signaling [55].

In terms of genetic modifiers, those with the PNPLA3 polymorphism show increased vulnerability to steatohepatitis and fibrosis, as well as a more than threefold increased chance of developing HCC [56,57]. In conjunction with the inherent genetic predisposition, the accumulation of lipids in the liver results in metabolic reprogramming, which is marked by a confluence of cellular and metabolic modifications, as well as by the buildup of metabolites with harmful potential [58]. The combination of an inflammatory microenvironment with abnormal metabolism and continuous liver regeneration represents a major contributing factor in DNA instability and liver tumorigenesis among MAFLD individuals [58] (Figure 2).

In more detail, the immune system plays a crucial role in the setting of MAFLD and HCC since the presence of an inflammatory reaction in the liver is a defining characteristic of MASH and is believed to be the primary factor driving disease progression towards fibrosis, cirrhosis, and/or HCC [59]. Multiple studies have provided evidence for the significant involvement of innate and adaptive immune pathways in generating hepatic inflammation in MASH, while many preclinical models have established the relevance of fibrosis and the immune response in the progression of MASH-associated HCC [60]. In the context of MASH, recent data have shown the influence of both adaptive and innate immune cells on the liver microenvironment, specifically in relation to the transition from MASH to HCC [61,62]. These immune cells include CD4+ T cells, metabolically activated CD8+ T cells, platelets, and dendritic cells. Both animal and human studies have demonstrated that the hepatic population of CD8+PD1+ T cells exhibited an upward trend as the pathogenesis of MASH advanced [63,64]. The aforementioned cells exhibit a reiteration of an auto-aggressive condition, whereby liver-resident CD8+PD1+CD103+ T cells, although exhausted, demonstrate an activated phenotype by producing elevated quantities of cytokines such as TNF-α, CCL2, IL-10, or granzyme B [63,64]. The inflammation-induced suppression of cytotoxic CD8+ T lymphocyte activation may consist of a tumor-promoting mechanism [65]. This immune dysfunction in the liver microenvironment, in conjunction with gut dysbiosis and the apparent role of progressive fibrosis, constitutes the complex pathogenesis of HCC among MAFLD patients, with an annual incidence rate of 2.5–13% depending on the disease stage [65,66].

## 4. Obesity and HCC: Molecular Pathways and Pathogenesis

Approximately 20% to 30% of MAFLD-related HCCs are developed de novo in the absence of cirrhosis [16], indicating the presence of distinct pathophysiologic mechanisms differentiating it from cirrhosis-driven carcinogenesis (Table 1). Adipose tissue consists of two main compartments: the subcutaneous fat tissue and the intra-abdominal (omental, mesenteric, and perirenal) or visceral fat deposits with several functional differences [67]. In fact, the omentum is an immunological organ within the peritoneal cavity harboring organized leukocyte aggregates known as “milky spots” and fat-associated lymphoid clusters (FALCs) [68], being primarily responsible for obesity-related inflammation. Its anatomic proximity to the liver, since omental and mesenteric fat deposits are drained by the portal vein, while rectal and perirenal fat is drained directly into the systemic circulation, might imply a functional interconnection as stated by the “portal theory” [67], which states that the liver is exposed in greater concentrations to pro-inflammatory and FFAs from the intra-abdominal fat storages, predisposing patients to hepatic steatosis and IR [67], driving hepatocarcinogenesis. Obesity exerts a multitude of effects both locally and systematically, through a series of inflammatory mediators (increased leptin, reduced adiponectin, TNF-α, IL-6), growth factors (insulin, IGF-1), and metabolism molecules (visfatin, grehlin, and resistin) [69]. Cytokines produced by adipose tissue include IL-6, IL-8, IL-1β, TNF-α, vascular endothelial growth factor (VEGF), chemokine (C-C motif) ligand 2 (CCL2), and CCL5, which, apart from recruiting immune cells, also support neovascularization [70] (Figure 3).

With respect to circulating inflammatory cells of the bloodstream, obesity is characterized by downregulation of CD8+ T cells and Tregs, with concomitant induction of M1 macrophages, CD4+ T cells, B cells, and NK cells [69,71], having an impact on the HCC’s tumor microenvironment (TME) [69,72,73,74,75]. Preclinical studies have documented that leptin enhances neovascularization through a VEGF-dependent pathway in parallel with the induction of liver fibrosis and hepatocarcinogenesis [76]. In contrast, adiponectin reduces tumor growth, HSC, and macrophage activation in orthotopic animal tumor models, downregulating angiogenesis through the inhibition of Rho kinase (ROCK)/IFN-inducible protein 10 (IP10)/vascular endothelial growth factor (VEGF) [77], while animal MASH models fed choline-deficient l-amino acid-defined (CDAA) diets exhibited hypoadiponectinemia which further aggravated fibrogenesis, inflammation and hepatocarcinogenesis [78]. The induction of lipogenic signaling is associated with tumor progression and poor prognosis [79]. Low-grade obesity-induced inflammation leads to the generation of IL-6 and TNF-α with tumor-promoting effects, activating tumor-associated neutrophils and macrophages [80]. IL-6 further induces the activation of STAT3, orchestrating hepatic carcinogenesis [81,82].

Chronic oxidative stress contributes to hepatocarcinogenesis, regulating the expression of STAT1 and STAT3 [83]. In more detail, the obesity-induced oxidative environment leads to the inactivation of the T cell protein tyrosine phosphatase (TCPTP), upregulating STAT1 and STAT3 signaling [84]. The unopposed STAT1 signaling facilitates lymphocyte recruitment, causing MASH and fibrosis, while STAT3 signaling promotes the development of HCC [84]. In addition, Omega-3 PUFAs downregulate the expression of STAT3-inducing apoptosis, comprising a potential therapeutic target [85]. Moreover, the activation of tumor necrosis factor receptor (TNFR) signaling by TNF-α leads to the stimulation of c-Jun N-terminal kinase (JNK)/activator protein-1 (AP-1) signaling, potentiating hepatocellular proliferation and inhibiting apoptosis [73,86]. The latter is supported by clinical studies, which robustly demonstrate an essential role of TNF-α in hepatocarcinogenesis [87]. Conclusively, insulin resistance represents a major cause of carcinogenesis [73,88].

Hyperinsulinemia causes the activation of IGF-1 and insulin receptor substrate-1 (IRS-1), which further upregulate several signaling molecules, including p53, mitogen-activated protein kinase (MAPK) [89,90], and phosphatidylinositol-3 kinase (PI3K)/Akt [89,90], leading to cellular proliferation while also downregulating apoptosis [91]. Additionally, hyperglycemia triggers the upregulation of hypoxia-inducible factor-1α (HIF1α), leading to the induction of glycolytic enzymes, which provide a survival advantage under hypoxic conditions [92]. According to an observational study by Jee et al., an elevated fasting serum glucose comprises an independent risk factor of HCC mortality among men (HR 1.57, 95% CI 1.40–1.76, *p* = 0.03) and women (HR 1.33, 95% CI 1.01–1.81, *p* = 0.045), increasing HCC incidence rate in men (HR 1.72, 95% CI 1.56–1.89, *p* = 0.01) [93].

The HCC microenvironment has been vigorously studied over the past decade [94,95,96,97], comprising, apart from hepatocellular tumor cells, innate immune cells (macrophages [98], neutrophils [99], dendritic cells), lymphoid cells, adipocytes, stellate cells, fibroblasts, and ECM, and endothelial cells implicated in a complex interplay to overcome the Warburg effect-induced hypoxic and acidic conditions [100,101]. A growing body of evidence suggests that obesity shapes the liver microenvironment, promoting carcinogenesis [69]. Fundamental differences are found in the immunologic synthesis of the omentum and liver among lean and obese individuals. The omentum and liver are tolerogenic organs, which is achieved by the predominance of anti-inflammatory innate and adaptive cell populations producing IL10. In the omentum predominate M2 macrophages, Tregs, invariant natural killer T (iNKT) cells and Th2 CD4+ (Th2) cells [68], while in the liver there is an abundance of Tregs in conjunction with NK cells, NKT cells, mucosal-associated invariant T (MAIT) cells, γδ T cells, CD4+ and CD8+ T cells and B cells [69]. The accumulation of lipids within adipose tissue and hepatocytes shifts the balance towards pro-inflammatory cellular populations such as Th17 CD4+ and CD8+ T cells, NK cells, B cells, M1 macrophages and neutrophils in omentum [68], and CD14+ Kupffer cells [102], MOP+ neutrophils [103], TNF+ dendritic cells [104], TH17 CD4+ T cells [105], CD8+ T cells [106], and CD3+CD56+ Vα24+ iNKT cells [107].

Although CD8+ T cells and NK cells exhibit cancer immunosurveillance properties, recent evidence from various studies suggested that obesity induces functional alterations leading to immunoparesis [108,109,110,111]. CD8+ T cells, in contrast to cancer cells, lack the ability for metabolic adaptations since they are unable to increase their fat uptake upon exposure to a high-fat diet (HFD) [108] and are sensitive to nutrient deprivation such as glutamine-expressing exhaustion phenotypes [110]. Targeting cancer cells’ metabolic adaptation mechanisms could offer a potential therapeutic strategy [108]. Analogously, the upregulation of sterol regulatory element-binding protein 2 (SREBP2) leads to an intracellular increase in cholesterol and lipid peroxide, causing the suppression of natural killer T (NKT) cell cytotoxicity [111]. In regard to liver-resident macrophages, hypernutrition leads to defective macrophage cleavage, while the upregulation of IL-1β and TNF potentiates an ADAM-17-mediated proteolytic cleavage of the macrophage phagocytic receptor TREM2, resulting in the aggregation of damaged hepatocytes, stimulating further the inflammatory process [112].

According to Hanahan et al., inflammation has been recognized as a tumor-promoting characteristic [94,113], and besides their immunoregulatory functions, inflammatory cells exert pleiotropic tumor-promoting functions within the HCC TME. HepG2 cells co-cultured with innate and adaptive cells exhibited epithelial-to-mesenchymal transition (EMT) upon inflammatory stimulation, which promoted immune escape and metastatic dissemination [114]. Additionally, HCV-carrier mice fed with high cholesterol and saturated fat diet (HCFD) upon TLR4-stimulation developed HCC with more prominent mesenchymal features, highly suggestive of activated EMT [115]. Finally, evidence about the role of the IL-6/STAT3 axis in the organization of HCC TME has emerged [116,117]. Zheng et al. demonstrated that the upregulation of IL-6/STAT3 signaling in Huh7 cells generated the transformation of normal fibroblasts to cancer-associated fibroblasts (CAFs), potentiating the expression of TIMP-1. Consequently, CAFs induced the proliferation of Huh7 cells, generating a positive feedback loop [116]. Additionally, CAFs induced EMT in HCC, activating the IL-6/STAT3 signaling [117]. Thus, it becomes evident that obesity modulation of HCC TME exerts pleiotropic effects in the generation and the progression of hepatocarcinogenesis. Further exploitation of the steatosis-driven tumor microenvironment of HCC could offer unique personalized theranostic applications [118].

**Table 1 ijms-24-14704-t001:** Summary of studies evaluating the role of obesity in HCC pathogenesis.

Study (Year)	Study Subjects	Outcomes
Aleffi et al. [70] (2005)	Human	Pro-inflammatory cytokines produced by adipose tissue include IL-6, IL-12, TNF-α, VEGF, CCL2, and CCL5, which regulate immunity and promote neovascularization in human hepatic stellate cells, mediated via the activation of NF-κB
Endo et al. [54] (2006)	Animal	The presence of dysfunctional adipose tissue characterized by reduced adiponectin and elevated leptin secretion plays a crucial role in promoting tumor proliferation and angiogenesis, mediated by OB-R/STAT3 signaling
Kitade et al. [76] (2006)	Animal	Leptin-mediated neovascularization coordinated with VEGF plays an important role in the development of liver fibrosis and hepatocarcinogenesis in MASH
Man et al. [77] (2010)	Animal	Adiponectin reduces HCC growth, hepatic stellate cell and macrophage activation, downregulating angiogenesis via the inhibition of ROCK/CXCL10/VEGF
Park et al. [80] (2010)	Animal	Obesity-promoted HCC development is dependent on enhanced production of the tumor-promoting cytokines IL-6 and TNF, which cause hepatic inflammation and activation of the oncogenic transcription factor STAT3
Grohmann et al. [84] (2008)	Animal	Obesity-associated hepatic oxidative stress can independently contribute to the pathogenesis of MASH, fibrosis, and HCC by the inactivation of TCPTP and the upregulation of STAT1 and STAT3 signaling
Rensen et al. [103] (2023)	Human	Increased hepatic myeloperoxidase activity in obese subjects is associated with the induction of CXC chemokines and hepatic neutrophil infiltration, contributing to the development of HCC
Eferl et al. [86] (2003)	Animal	Activation of TNFR signaling by TNF-α leads to the stimulation of JNK/AP-1 signaling, potentiating hepatocellular proliferation and inhibiting apoptosis
Wang et al. [112] (2022)	Animal	Obesity leads to defective macrophage cleavage, while the upregulation of IL-1β and TNF potentiates a TACE-mediated proteolytic cleavage of TREM2, resulting in the aggregation of damaged hepatocytes, stimulating the inflammatory process
Kumar et al. [115] (2006)	Animal	HCFD stimulates the TLR4 and the OB-R/phosphoSTAT3 signaling pathways, resulting in liver tumorigenesis via an exaggerated mesenchymal phenotype with prominent Twist1-expressing TICs

CCL: chemokine (C-C motif) ligand; NF-κB: nuclear factor kappa-light-chain-enhancer of activated B cells; CXCL10: CXC motif chemokine ligand 10; ROCK: Rho-kinase; VEGF: vascular endothelial growth factor; STAT: signal transducer and activator of transcription; TCPTP: T cell protein tyrosine phosphatase; HCC: hepatocellular carcinoma; MASH: metabolic associated steatohepatitis; AP-1: activator protein-1; JNK: c-Jun N-terminal Kinase; TNFR: tumor necrosis factor receptor; TREM2: triggering receptor expressed on myeloid cells 2; TACE: tumor necrosis factor-α-converting enzyme; TNF: tumor necrosis factor; IL: interleukin; OB-R: leptin receptor; TRL4: Toll-like receptor-4; TIC tumor-initiating stem-like cell; HCFD: high in cholesterol and saturated fat diet.

## 5. Treatment Strategies in Obesity-Related HCC

Treatment options for patients with HCC were thoroughly presented in the European and American guidelines for therapeutic management, which demonstrated that treatment selection is primarily based on disease stage [16,119,120]. The most extensively used staging approach for HCC is the Barcelona Clinic of Liver Cancer (BCLC) algorithm, which distinguishes patients with HCC into five clinical stages: very early stage (BCLC 0), early stage (BCLC A), intermediate stage (BCLC B), advanced stage (BCLC C), and terminal stage (BCLC D) [16,121]. Nevertheless, current guidelines do not distinguish between obesity/MAFLD-related HCC and HCC associated with other modalities, e.g., chronic viral hepatitis [65]. Patients with obesity-related HCC usually present with other obesity-associated cardiometabolic comorbidities, affecting their general health status and influencing therapeutic decision-making [65,122,123,124,125]. These patients are usually diagnosed at an older age at stages BCLC C or BCLC D, even though without the obvious manifestations of cirrhosis [65,126]. The latter is mainly attributed to low-performance status test (PST), influenced by the coexisting cardiometabolic diseases [65].

As in every other obesity-related comorbidity or complication, prevention of disease progression via the achievement of weight loss is of paramount importance [127,128]. The armory against obesity consists of lifestyle modifications, such as dietary and exercise interventions, anti-obesity, anti-diabetic, and anti-hyperlipidemic medications, bariatric/metabolic surgery, and anti-inflammatory drugs [128,129,130]. Lifestyle changes focusing on weight loss are cardinal for reducing the risk of progression of obesity to MAFLD to MASH; however, dietary guidance as part of supportive care for patients with HCC should also aim to maintain a well-balanced diet with sufficient caloric intake to counteract weight loss or malnutrition, which are frequently seen in patients with cancer [65]. Several studies have shown favorable results with the optimal control of metabolic conditions, primarily T2DM. Patients with cirrhosis receiving metformin for T2DM had a lower incidence of HCC [131]. In a meta-analysis of patients with T2DM and HCC who were receiving anti-diabetic medication, metformin was associated with increased overall survival [132]. However, these results cannot be generalized as it appears that racial/ethnic disparity exists with the use of metformin [133].

Another example is the relatively new class of anti-diabetic drugs, sodium-glucose cotransporter-2 inhibitors (SGLT2i), which promote weight loss [124], are effective against NAFLD and have recently shown promising results in individuals with T2DM and HCC [134]. In addition to improving metabolic markers, such as glycated hemoglobin and BMI, the possible benefits of SGLT2i in HCC could be mediated by the wealth of their pleiotropic actions, as these drugs are known to alleviate systemic inflammation, improve cell bioenergetics, and down-regulate oxidative stress [135]. Glucagon-like peptide 1 (GLP-1) receptor agonists (GLP-1 RA) and the dual GLP-1/gastric inhibitory polypeptide agonist tirzepatide are new anti-diabetic agents that induce significant reductions in body weight and have been shown to improve hepatic biochemistry and reduce liver fat content in patients with MAFLD [136,137]. GLP-1 RA has shown encouraging results in animal models with HCC [138]. However, it is still under investigation whether these benefits are driven by their metabolic effects or whether their anti-inflammatory properties are also involved.

Bariatric surgery is an ideal option for patients with obesity and/or MASH who are unresponsive to other treatments, while it has been demonstrated to both ameliorate hallmarks of MASH as well as to decrease the risk of developing HCC [139,140,141,142]. The use of statins (for hypercholesterolemic states usually accompanying obesity and T2DM) has been shown to potentially exert beneficial inhibitory effects on HCC development and HCC recurrence after curative resection [142,143]. Moreover, supplementation of aspirin was shown to reduce the risk of progression to advanced fibrosis, and subsequently HCC, in patients with MAFLD and MASH [144,145]. These findings were attributed to the anti-inflammatory effects of aspirin, which could be of aid in obesity, as it is associated with chronic low-grade inflammation [122,145]. All of the aforementioned findings should be interpreted with caution, given the paucity of randomized human trials, since a significant body of evidence comes from animal and observational studies that are prone to bias. Furthermore, these strategies need further investigation on top of the standard treatments already in use to tackle metabolic-associated HCC.

Surgery is the main curative treatment option for localized HCC, including both liver resection and/or liver transplantation for those with multifocal disease and without cirrhosis, as the presence of the latter complicates the management [16,146]. Sparce studies have explored the influence of obesity on HCC peri/post-surgically. A recent multicenter study examined the coexistence of HBV infection and obesity as part of metabolic syndrome and has demonstrated that metabolic syndrome was associated with poor prognosis, including many postoperative long-term oncologic survival indices [147]. However, it has not been established yet whether metabolic treatments can directly benefit these individuals. Other local treatment options, curative or supportive, include thermal ablation, intra-arterial therapies (particle embolization, radioembolization, or drug-eluting bead), and radiotherapy [16,148]. For later stages of HCC, systemic drug administration is the treatment of choice [16,65,121]. Sorafenib, a tyrosine kinase inhibitor, was the drug of choice for HCC for several years [16,65,149], but various other novel therapeutic options have been approved [16,65,119,150,151]. Overall, upon evaluating and deciding on available treatment options for patients with HCC, a thorough assessment by multidisciplinary teams in experienced centers should be performed, and the most suitable therapy for each patient should be considered via constant re-evaluation [16,65].

## 6. Discussion

Understanding the molecular pathways that link obesity and HCC is crucial for the development of targeted therapies and preventive strategies. The molecular linkage between HCC and obesity is multifaceted, involving inflammation, insulin resistance, dyslipidemia, gut microbiota, and adipokines. A comprehensive understanding of these pathways provides a foundation for the development of targeted therapies and preventive measures that can potentially reduce the incidence and impact of HCC in the context of obesity. Continued research efforts are essential to unravel the intricacies of this complex relationship and translate findings into clinical practice. Inflammation and immune dysregulation are the hallmarks of obesity-related HCC. Obesity is characterized by chronic low-grade inflammation, mainly mediated by adipose tissue-derived cytokines such as TNF-α and IL-6 [152]. These inflammatory mediators create a microenvironment in the liver promoting hepatocyte damage and regeneration, increasing the risk of genetic mutations that drive HCC. The NF-κB pathway, a central regulator of inflammation, plays a pivotal role in connecting obesity-induced inflammation with HCC development. Activation of NF-κB promotes the production of pro-inflammatory cytokines and induces cell survival pathways, ultimately contributing to HCC initiation and progression [153]. Moreover, obesity is closely linked to insulin resistance and hyperinsulinemia, which lead to increased IGF-1 levels, activating the PI3K/AKT/mTOR pathway, which is frequently dysregulated in HCC [154]. This pathway promotes cell proliferation and inhibits apoptosis, fostering HCC development. Additionally, hyperinsulinemia can stimulate HSCs, leading to liver fibrosis, a precursor condition for HCC. Furthermore, dyslipidemia often accompanies obesity and can result in the accumulation of lipids in hepatocytes, which can progress to MASH, liver cirrhosis, and eventually HCC. Peroxisome proliferator-activated receptor gamma (PPAR-γ) and sterol regulatory element-binding protein-1c (SREBP-1c) are key transcription factors that regulate lipid metabolism and are implicated in the link between obesity and HCC [155]. Activation of these pathways leads to increased lipid synthesis and storage, contributing to liver carcinogenesis. Adipose tissue secretes various adipokines, including leptin and adiponectin, which play roles in energy homeostasis and inflammation. Elevated levels of leptin and reduced adiponectin are common in obesity and have been associated with HCC development [156]. Leptin promotes cell proliferation and angiogenesis, while adiponectin has anti-inflammatory and anti-tumor properties. The imbalance in these adipokines contributes to the molecular linkage between obesity and HCC. Finally, obesity-associated changes in the gut microbiota composition can promote the release of pro-inflammatory LPS into the bloodstream. LPS-induced inflammation can activate TLR4 signaling, triggering downstream pathways like NF-κB, which exacerbate liver inflammation and fibrosis, ultimately increasing the risk of HCC [157].

Non-invasive imaging techniques and biomarkers of MAFLD could aid in both the prevention of HCC as well as in timely diagnosis and therapy [158]. Moreover, developing reliable biomarkers for early HCC detection in obese populations is of utmost importance. Alpha-fetoprotein (AFP), the first serum biomarker of HCC, was discovered in 1963, and it has been demonstrated that it activates AKT/mTOR signaling to promote CXCR4 expression and migration of hepatoma cells, while several other protein biomarkers have been identified and utilized into clinical practice since then [159]. Yet, insufficient specificity and sensitivity of these biomarkers highlight the need for novel biomarker discovery. Current integrated multiomics technologies for the identification of gene expression and protein or metabolite distribution patterns can facilitate the discovery of novel biomarker candidates for obesity-related HCC early diagnosis and prognosis [160]. Nowadays, the biomarkers for combined testing for HCC diagnosis, tumor recurrence, and treatment response include the “gold standard” serum biomarker AFP, as well as the serum biomarker lens culinaris agglutinin (LCA)-reactive L3 glycoform of AFP (AFP-L3), which has been associated with poor prognosis upon high concentrations for patients with HCC [161], and des-gammacarboxyprothrombin (DCP), which stimulates HCC invasion and angiogenesis through activation of matrix metalloproteinase, via the upregulation of extracellular signal-regulated kinase-mitogen-activated protein kinase (MAPK) and the DCP-kinase insert domain receptor-phospholipase Cγ-MAPK pathways [162]. Another promising biomarker includes osteopontin (OPN), an acidic chemokine-like secreted phosphoglycoprotein found in ECM that functions via integrin-αv3/NF-κB/HIF-α, PI3K/Akt/NF-κB, and CD44-mediated signaling and can be used for early diagnosis of HCC [163,164]. Moreover, glypican-3 (GCP3) is a heparan sulfate proteoglycan that can stimulate HCC growth through canonical Wnt/β-catenin-mediated signaling pathway, and its combination with AFP improves the sensitivity for HCC diagnosis up to 82% or 94%, depending on the HCC type [165]. Elevated midkine (MDK) levels have been observed both in tumor biopsies and in the blood serum of HCC patients; thus, it can serve as a biomarker of HCC progression and metastasis via ERK/JNK/p38 MAPK-mediated signaling promoted by ZFAS1, which is elevated in HCC but can be suppressed using microRNA-624 [166]. It is of note that the sensitivity of MDK for HCC diagnosis is higher than that of AFP, even at the early stage of HCC [167]. Similarly, dickkopf-1 protein (DKK1) can serve as a biomarker of HCC, contributing to the identification of patients with AFP-negative HCC and acting as an inhibitor of Wnt/β-catenin signaling [168]. DKK1 promotes inflammation, cell migration, and invasion in HCC via TGF-1-mediated remodeling of the TME, while it exerts oncogenic effects in HepG2/C3C cell lines and thus is a promising target for HCC immunotherapy [169]. Finally, squamous cell carcinoma antigen (SCCA) or SCCA-immunoglobulin M (IgM) have demonstrated moderate accuracy in HCC diagnosis; however, combined measurements with AFP and DCP increase the sensitivity, specificity, and diagnostic accuracy for HCC [170,171].

Understanding the intricate molecular pathways linking obesity and HCC has opened avenues for potential therapeutic interventions and preventive strategies. Developing drugs that specifically target key molecules in the molecular pathways connecting obesity and HCC, such as inhibitors of NF-κB, PI3K/AKT/mTOR, and PPAR-γ, may prove effective in preventing or treating HCC in obese individuals. Lifestyle modifications, including weight loss through diet and exercise, remain a cornerstone in reducing HCC risk in obese individuals by helping to alleviate insulin resistance, inflammation, and dyslipidemia. In addition, research into the manipulation of gut microbiota composition to mitigate inflammation and reduce HCC risk holds promise, while probiotics, prebiotics, and dietary interventions may also be explored. Finally, promoting obesity prevention and management on a population scale is crucial, while public health campaigns emphasizing the importance of maintaining a healthy weight and lifestyle can contribute to reducing the burden of obesity-related HCC.

## 7. Conclusions

In conclusion, this comprehensive review delved into the intricate relationship between obesity and HCC, shedding light on the multifaceted mechanisms underlying their association underscoring the substantial role that obesity plays in the development, progression, and poor prognosis of HCC. Through the elucidation of various molecular, metabolic, and inflammatory pathways, we have provided insights into how adipose tissue-derived factors, altered insulin signaling, chronic inflammation, and dysregulated lipid metabolism collectively contribute to the increased risk of HCC among individuals with obesity. However, despite the significant strides made in understanding this relationship, there remain several gaps in knowledge that merit further investigation. The need for future trials and studies is evident in order to establish a more comprehensive understanding of the causal links between obesity and HCC, while prospective cohort studies with larger and more diverse populations are essential to unravel the intricate aspects of this relationship. Additionally, molecular studies that delve into specific pathways and potential therapeutic targets can offer novel interventions to mitigate the risk and progression of HCC among individuals with obesity. Incorporating interdisciplinary approaches, such as combining clinical, molecular, and epidemiological insights, will be pivotal in deciphering the intricate interplay between obesity and HCC. By adopting a holistic perspective and focusing on the diverse factors at play, future trials can pave the way for personalized prevention strategies and targeted treatments that address both obesity and HCC simultaneously.

## Figures and Tables

**Figure 1 ijms-24-14704-f001:**
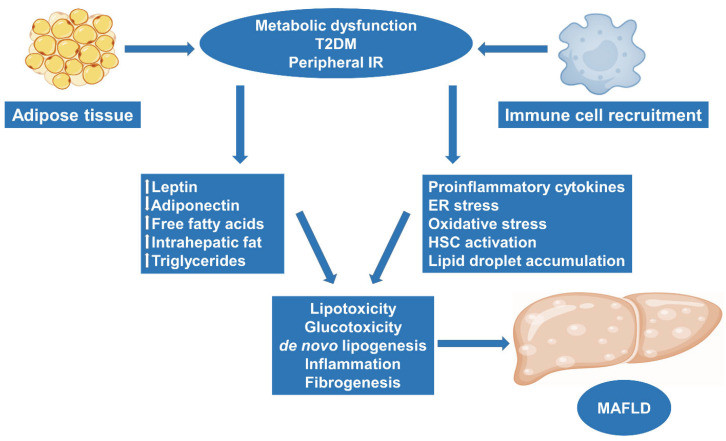
Pathogenesis of MAFLD as a continuum from obesity to metabolic dysregulation and diabetes. Obesity leads to adipocyte and metabolic dysfunction, peripheral tissue insulin resistance, and increases the risk for T2DM. In this setting, circulating free fatty acids and leptin increase while adiponectin decreases, resulting in increased triglyceride and intrahepatic fat accumulation, further potentiating de novo lipogenesis and lipotoxicity with increased oxidative and ER stress, leading to a microenvironment favoring MAFLD. Adipose tissue is further infiltrated by pro-inflammatory cytokines and interleukins, contributing to chronic low-grade intrahepatic inflammation characterized by increased HSC activation, and fibrogenesis ensues. T2DM: type 2 diabetes mellitus; IR: insulin resistance; HSC: hepatic stellate cell; ER: endoplasmic reticulum; MAFLD: metabolic dysfunction-associated fatty liver disease.

**Figure 2 ijms-24-14704-f002:**
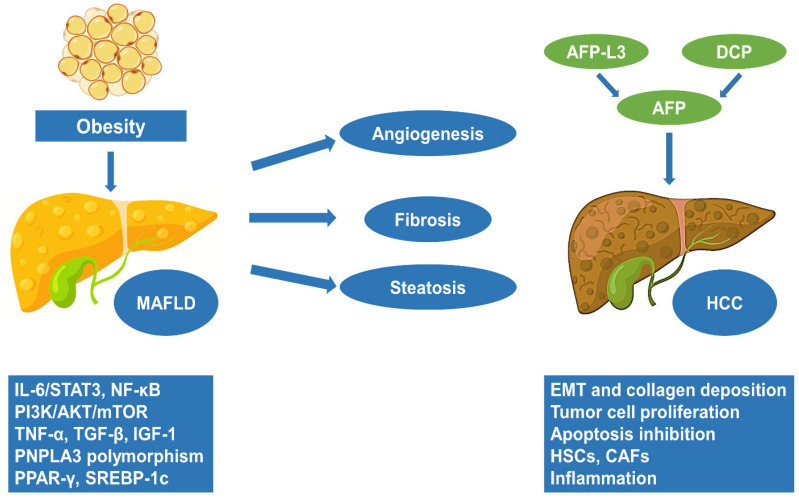
Pathogenesis of HCC as a continuum from MAFLD. The complex association between MAFLD and HCC development is mediated by the interaction of multiple pathogenic pathways, creating a microenvironment favoring carcinogenesis, characterized by chronic low-grade inflammation in the setting of steatohepatitis, increased EMT and fibrosis, neoangiogenesis and increased tumor cell proliferation. Therefore, using reliable biomarkers that provide aid in the early diagnosis of HCC, tumor recurrence, and response to therapy is of utmost importance for individuals with MAFLD. MAFLD: metabolic dysfunction-associated fatty liver disease; HCC: hepatocellular carcinoma; EMT: epithelial-to-mesenchymal transition; AFP: alpha-fetoprotein; DCP: des-γ-carboxy prothrombin; AFP-L3: lectin-reactive alpha-fetoprotein; IL: interleukin; IGF: insulin-like growth factor; STAT: signal transducer and activator of transcription; NF-κB: nuclear factor kappa-light-chain-enhancer of activated B cells; P13K: phosphoinositide 3-kinase; TNF: tumor necrosis factor; TGF: transforming growth factor; PNPLA: patatin-like phospholipase domain-containing protein; PPAR: peroxisome proliferator-activated receptors; SREBP: sterol regulatory element binding proteins; mTOR: mammalian target of rapamycin; CAF: cancer-associated fibroblasts HSC: hepatic stellate cell.

**Figure 3 ijms-24-14704-f003:**
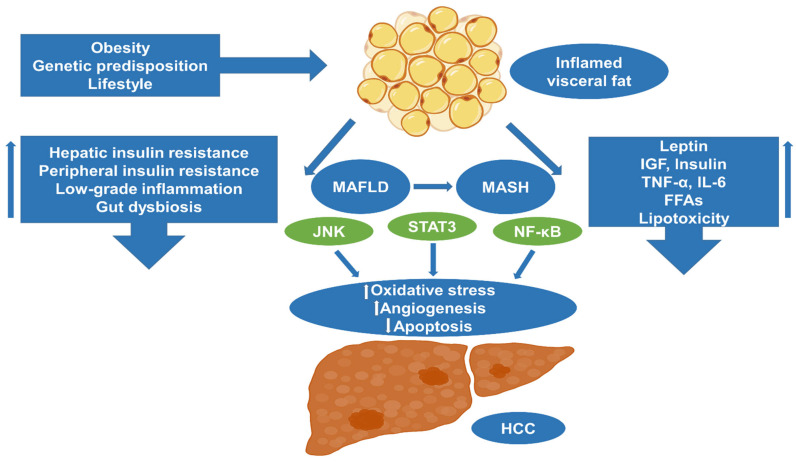
Pathogenic mechanisms involved in the development of HCC in the presence of obesity. An excessive influx of fatty acids into hepatocytes contributes to the promotion of oxidative stress, which, in turn, leads to the development of insulin resistance and hinders the process of autophagy, resulting in the activation of apoptotic cascades, eventually leading to tissue damage and inflammation. Chronic oxidative stress and IL-6 upregulate STAT3 and NF-κB, while enhanced TNF-α signaling activates the JNK pathway, potentiating hepatocellular proliferation and angiogenesis while inhibiting apoptosis. Chronic low-grade inflammation and insulin resistance through IGF-1 activation contribute to the creation of a microenvironment favoring HCC. In addition, the presence of dysfunctional adipose tissue characterized by reduced adiponectin and elevated leptin secretion plays a crucial role in promoting cellular proliferation and angiogenesis. Lastly, the presence of gut dysbiosis and increased intestinal permeability promotes tumorigenesis via enhanced IL-6 signaling. HCC: hepatocellular carcinoma; MAFLD: metabolic dysfunction-associated fatty liver disease; MASH: metabolic dysfunction-associated steatohepatitis; IGF: insulin-like growth factor; ROS: reactive oxygen species; TNF-α: tumor necrosis factor alpha; IL: interleukin; FFAs: free fatty acids; JNK: c-Jun N-terminal kinase; NF-κΒ: nuclear factor kappa-light-chain-enhancer of activated B cells; STAT3: signal transducer and activator of transcription 3.

## Data Availability

Not applicable.

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
