# Peer review of "Meeting at the Crossroad between Obesity and Hepatic Carcinogenesis: Unique Pathophysiological Pathways Raise Expectations for Innovative Therapeutic Approaches"

_ijms, 2023, doi:10.3390/ijms241914704_

Round 1

Reviewer 1 Report

The submitted manuscript entitled “Meeting at the Crossroad Between Obesity and Hepatic Carcinogenesis: Unique Pathophysiological Pathways Raise Expectations for Innovative Therapeutic Approaches” focuses on discussion relationships between obesity and hepatocellular carcinoma and the mechanisms underlying their association. This study scientifically sounds and may be of interest for the journal audience. The manuscript contains 137 references and 69 of them were published last 5 year. There are 1 Table and 1 Figure are presented to illustrate the topic. However, there are some concerns and recommendations to improve the quality of the manuscript. There are as follows:

Major concern:

Generally, the review lacks the wholeness and contains fragmented data with no concept or novel insight. Since the authors stated in the Abstract that they aimed to discuss the “landscape of molecular carcinogenesis in the context of obesity-driven HCC” it is recommended to provide more profound discussion on molecular basis of linkages between HCC and obesity, molecular biomarkers of both pathologies. The authors are recommended to read and cite the following papers:  doi: 10.3390/biomedicines9020159, doi: 10.14218/JCTH.2021.00248. etc. 

Other concerns:

1.     In the Introduction, in general MAFLD is poorly characterized; it would be useful to give short characteristics of MAFLD including difference from NAFLD. See doi: 10.3350/cmh.2022.0367

2.     Section 2, paragraph 1, Ref [29] does not support the authors’ statement. Instead, the following paper can be cited: doi: 10.3350/cmh.2022.0367.

3.     Section 2, paragraph 2, why MASH is important for this review? If so, this should be more clearly discussed.

4.     Sections 2 and 4, schematic representation of relationship between the pathways would be useful.

5.     Section 4, p.5, last paragraph, what is “circulating inflammatory cells’?

6.     Figure 1 should explain how STAT2, JNK, and NF-kB influence the pathways – this should be indicated using arrows and explanation in the Fig. legend.

7.     Explain the abbreviations including PNPLA3, NASH, etc.

8.     English grammar and style should be improved.

Editing is needed

Author Response

First Department of Internal Medicine

AHEPA University Hospital,  

St. Kyriakidi 1, 54636

Aristotle University of Thessaloniki,

Thessaloniki, Greece.

September 21st, 2023

RE: Meeting at the Crossroad Between Obesity and Hepatic Carcinogenesis: Unique Pathophysiological Pathways Raise Expectations for Innovative Therapeutic Approaches

We thank you for carefully evaluating our manuscript and for the positive and constructive feedback. Hopefully, our responses below address the points raised and pave the way for successfully publishing our work in your esteemed journal. All changes made are presented with track changes in the revised manuscript as a Word file.

Reviewer 1

Point 1. The submitted manuscript entitled “Meeting at the Crossroad Between Obesity and Hepatic Carcinogenesis: Unique Pathophysiological Pathways Raise Expectations for Innovative Therapeutic Approaches” focuses on discussing the relationship between obesity and hepatocellular carcinoma and the mechanisms underlying their association. This study is scientifically sound and may be of interest for the journal’s audience. The manuscript contains 137 references and 69 of them were published within the last 5 years. There are 1 Table and 1 Figure illustrating the topic.

Response: We thank the reviewer for the positive comments. We hope that the revised version of our manuscript, meets the reviewer’s high standards and requirements for publication, as we have implemented all of his/her suggestions in order to improve the overall quality of our paper.

Point 2. However, there are some concerns and recommendations to improve the quality of the manuscript. Generally, the review lacks the wholeness and contains fragmented data with no concept or novel insight. Since the authors stated in the Abstract that they aimed to discuss the “landscape of molecular carcinogenesis in the context of obesity-driven HCC” it is recommended to provide more profound discussion on molecular basis of linkages between HCC and obesity, molecular biomarkers of both pathologies. The authors are recommended to read and cite the following papers:  doi: 10.3390/biomedicines9020159, doi: 10.14218/JCTH.2021.00248. etc.

Response: We thank the reviewer for the constructive criticism and suggestions that will improve the overall quality of our paper. We have therefore added a Discussion section before Conclusions, analyzing the molecular basis of linkages between HCC and obesity, while also citing both of the recommended papers (References 158 and 160 in the revised manuscript).

Point 3. In the Introduction, in general MAFLD is poorly characterized; it would be useful to give short characteristics of MAFLD including difference from NAFLD. See doi: 10.3350/cmh.2022.0367

Response: We thank the reviewer for the constructive criticism. We have modified the Introduction section accordingly, explaining key characteristics of MAFLD and the principal differences from NAFLD. We have also cited the article suggested by the reviewer (Reference no 15).

Point 4. Section 2, paragraph 1, Ref [29] does not support the authors’ statement. Instead, the following paper can be cited: doi: 10.3350/cmh.2022.0367.

Response: The aforementioned reference has been removed and the text has been modified accordingly, citing the recommended paper (Reference no 15 in the revised manuscript).

Point 5. Section 2, paragraph 2, why MASH is important for this review? If so, this should be more clearly discussed.

Response: We thank the reviewer for the constructive criticism. We thereby have added a paragraph in Section 2, explaining the importance of MASH, as well as the pathophysiological mechanisms that lead to its development.

Point 6. Sections 2 and 4, schematic representation of relationship between the pathways would be useful.

Response: We thank the reviewer for the comment. Ιn the revised version of the manuscript, each of the sections 2, 3, 4 contain a figure, accurately depicting their respective contents.

Point 7. Section 4, p.5, last paragraph, what is “circulating inflammatory cells”?

Response: We thank the reviewer for the query. Circulating inflammatory cells are inflammatory cells that exist and circulate in the blood flow and are measured by serologic tests, such as platelets, neutrophils, lymphocytes, monocytes, inflammatory cytokines and acute phase proteins. Circulating inflammatory cells in the tumor microenvironment are generally characterized by either pro-tumor or anti-tumor activity. We have edited the text so as the term is clearer for the readers. As we explain in the revised text, the circulating inflammatory cells in our instance are CD8+ T cells and Tregs, M1 macrophages, CD4+ T cells, B cells and natural killer cells: “In respect to circulating inflammatory cells of the bloodstream, obesity is characterized by a down-regulation of CD8+ T cells and Tregs, with concomitant induction of M1 macrophages, CD4+ T cells, B cells and NK cells [62],[64], having an impact on the HCC’s tumor microenvironment (TME)”.

Point 8. Figure 1 should explain how STAT3, JNK, and NF-kB influence the pathways – this should be indicated using arrows and explanation in the Fig. legend.

Response: We thank the reviewer for the suggestions. We have modified Figure 1 accordingly, STAT3, JNK, NF-kB now have arrows that point towards their effect, and the figure legend has been edited, providing a thorough explanation regarding the involved pathways.

Point 9. Explain the abbreviations including PNPLA3, NASH, etc.

Response: We thank the reviewer for the comment. After thoroughly editing the entire text, all of the abbreviations are now correctly explained in the revised version of the manuscript.

Point 10. English grammar and style should be improved.

Response: We thank the reviewer for the comment. English grammar and style editing has been applied throughout the manuscript.

Please also see the attachment included.

Reviewer 2 Report

Opinion about the manuscript (review) entitled “Meeting at the Crossroad Between Obesity and Hepatic Carcinogenesis: Unique Pathophysiological Pathways Raise Expectations for Innovative Therapeutic Approaches” sent to International Journal of Molecular Sciences (MDPI).

The assessed text touches on an extremely important topic that is very timely, while at the same time our knowledge in this area is changing dynamically through new discoveries. The idea of this review is commendable, but the execution leaves much to be desired.

Remarks:

One of the most significant shortcomings of this study, which is a review in itself, is the use of a number of other reviews as references. This conflicts with the very main idea of a review, which should gather the results of research described in scientific articles (like the latest ones) and not in other reviews.

Each of the paragraphs 2, 3, 4 should contain a separate graphic describing in detail respective mechanisms.

Example of drawback: in the paragraph 2: Obesity and MAFLD the authors cited 8 references. In my opinion the cited articles are not “new” enough and the three recent positions from 2020 and 2022 are all reviews. Such approach is not acceptable. Please make an effort and seek throughout the literature the existing articles describing experiments – not reviews.

The language is understandable.

Author Response

First Department of Internal Medicine

AHEPA University Hospital,  

St. Kyriakidi 1, 54636

Aristotle University of Thessaloniki,

Thessaloniki, Greece.

September 21st, 2023

RE: Meeting at the Crossroad Between Obesity and Hepatic Carcinogenesis: Unique Pathophysiological Pathways Raise Expectations for Innovative Therapeutic Approaches

We thank you for carefully evaluating our manuscript and for the positive and constructive feedback. Hopefully, our responses below address the points raised and pave the way for successfully publishing our work in your esteemed journal. All changes made are presented with track changes in the revised manuscript as a Word file.

Reviewer 2

Opinion about the manuscript (review) entitled “Meeting at the Crossroad Between Obesity and Hepatic Carcinogenesis: Unique Pathophysiological Pathways Raise Expectations for Innovative Therapeutic Approaches” sent to International Journal of Molecular Sciences (MDPI).

The assessed text touches on an extremely important topic that is very timely, while at the same time our knowledge in this area is changing dynamically through new discoveries. The idea of this review is commendable, but the execution leaves much to be desired.

Response: We thank the reviewer for the valuable comments. We hope that the revised version of our manuscript, meets the reviewer’s high standards and requirements for publication, as we have implemented all of his/her suggestions in order to improve the overall quality of our paper.

Point 1. One of the most significant shortcomings of this study, which is a review in itself, is the use of a number of other reviews as references. This conflicts with the very main idea of a review, which should gather the results of research described in scientific articles (like the latest ones) and not in other reviews.

Response: We thank the reviewer for the constructive criticism. We have modified the references in the revised version of the manuscript so that it cites less reviews and, instead, cites many more results from original articles.

Point 2. Each of the paragraphs 2, 3, 4 should contain a separate graphic describing in detail respective mechanisms.

Response: We thank the reviewer for the suggestion. In the revised version of the manuscript, each of the sections 2, 3, 4 contain a figure, accurately depicting their respective contents.

Point 3. Example of drawback: in the paragraph 2: Obesity and MAFLD the authors cited 8 references. In my opinion the cited articles are not “new” enough and the three recent positions from 2020 and 2022 are all reviews. Such approach is not acceptable. Please make an effort and seek throughout the literature the existing articles describing experiments – not reviews.

Response: We thank the reviewer for the constructive criticism and suggestions. In the revised version of the manuscript, and after also following the recommendations of Reviewer 1, in Section 2: Obesity and MAFLD, two more paragraphs have been added. There, we cite as many original studies as possible, while the previously cited references along with the respective text have been replaced with relevant “new” references, accordingly.

Point 4. The language is understandable.

Response: We thank the reviewer for the comment. English editing has been applied throughout the entire manuscript.

Please also see the attachment.

Reviewer 3 Report

It is a well-structured review. All the important areas have been covered with appropriate references.

Author Response

First Department of Internal Medicine

AHEPA University Hospital,  

St. Kyriakidi 1, 54636

Aristotle University of Thessaloniki,

Thessaloniki, Greece.

September 21st, 2023

RE: Meeting at the Crossroad Between Obesity and Hepatic Carcinogenesis: Unique Pathophysiological Pathways Raise Expectations for Innovative Therapeutic Approaches

We thank you for carefully evaluating our manuscript and for the positive and constructive feedback. Hopefully, our responses below address the points raised and pave the way for successfully publishing our work in your esteemed journal. All changes made are presented with track changes in the revised manuscript as a Word file.

Reviewer 3. It is a well-structured review. All the important areas have been covered with appropriate references.

Response: We thank the reviewer for the positive feedback and for taking the time to review our manuscript.

Please also see the attachment.

Round 2

Reviewer 1 Report

The authos have properly addressed my recommendations

Minor editing is required

Reviewer 2 Report

The revision is acceptable.